# Association of significant risk perception with the use of complementary and alternative medicine: A cross-sectional study in Hispanic patients with rheumatoid arthritis

Irazú Contreras-Yáñez[1], Ángel Cabrera-Vanegas[1], Andrea Robledo-Torres[1], Claudia Cáceres-Giles[1], Salvador Valverde-Hernández[1], Diana Padilla-Ortiz[2], Guillermo Arturo Guaracha-Basáñez[1], Virginia Pascual-Ramos[1] *

1 Immunology and Rheumatology Department, Instituto Nacional de Ciencias Médicas y Nutrición Salvador Zubirán, Mexico City, Mexico, 2 Spondyloarthropathies Research Group, Universidad de La Sabana, Hospital Militar Central, Bogotá DC, Colombia

⊚ These authors contributed equally to this work.

* virtichu@gmail.com

## Abstract

### Background

Risk perception (RP) describes patient´s judgment of the likelihood of experiencing something unpleasant, and has been associated to the adoption of health behaviors. Current rheumatoid arthritis (RA) guidelines recommend early and intensive treatment, although patients also commonly use Complementary and Alternative Medicine (CAM). We aimed to investigate if significant RP was associated to CAM use in Hispanic RA outpatients and to describe additional associated factors.

### Methods

Between March and August 2019, 266 consecutive outpatients were invited to a face-to-face interview to collect socio-demographic and RA-related information, to assess comorbidity and the following patient-reported-outcomes: pain, overall-disease and treatment adherence with visual analogue scales, disease activity with RAPID-3, RP with a validated questionnaire, and CAM use with a translated and cross-culturally adapted for Argentina version of the International CAM questionnaire. Medical records were reviewed to corroborate the data provided by the patients. CAM use definition was restricted to "in the last 3 months". Significant RP was defined based on published cut-off. Multiple logistic regression analysis was used to investigate factors associated to CAM use. The study received IRB approval.

### Results

There were 246 patients included, primarily middle-aged women, with substantial disease duration, moderate disease activity and 70 patients (28.5%) had significant RP. Two hundreds patients (81.3%) were CAM users. Significant RP (OR: 2.388, 95%CI: 1.044–5.464,

**Data Availability Statement:** All relevant data are within the paper and its Supporting Information files.

**Funding:** The authors received no specific funding for this work.

**Competing interests:** The authors have declared that no competing interest exist.

p = 0.039) and access to Federal health care system (OR: 2.916, 95%CI: 1.081–7.866, p = 0.035) were associated to CAM use.

## Conclusions

Patient´s perception of RA-related negative consequences was associated to recent CAM use in Hispanic RA outpatients.

## Introduction

Rheumatoid arthritis (RA) is a systemic inflammatory disorder with articular and extra-articular involvement that, if not properly controlled, can lead to significant structural damage, functional impairment, disability, reduced quality of life, and increased mortality [1–3]. Current treatment guidelines recommend that patients should adhere to a "treat-to-target" strategy using disease-modifying anti-rheumatic drugs (DMARDs), in order to achieve better outcomes [4, 5]. However, patients also commonly utilize alternative medicines to treat this disorder for varying reasons, including the chronic and painful nature of the disease, the lack of a known cure using traditional medicine, and the potential adverse events associated with DMARD use [6–21]. Meanwhile, nondisclosure rates of patients to their primary care physicians regarding the use of alternative medicines have been found to be in the range of 23% to 72% for non-rheumatologic patients [22] and 45% for rheumatologic patients [23].

The National Center for Complementary and Alternative Medicine of the United States National Institutes of Health defines complementary and alternative medicine (CAM) as "a group of diverse medical and health care systems, practices and products that are not presently considered to be part of conventional medicine" [24]. Meanwhile, the World Health Organization (WHO) adopts an anthropologic framework, and describes CAM as "a broad set of health care practices that are not part of the country´s own tradition and are not integrated into the dominant healthcare system" [25]. The lack of a consistent definition of CAM has extended to specific type of alternative therapies [22], although CAM modalities have traditionally been classified into the following five categories: biology-based therapies, manipulative and body-based practices, mind-body interventions, energy therapies, and whole medical systems [26]; importantly, whole medical systems include traditional Indian and Chinese medicine, as well as naturopathy; all three have been established as branches of mainstream medicine in India, China, and Germany/Central Europe, respectively, and therefore, may not be appropriately classified under the WHO definition of CAM, which hampers a uniform approach when analyzing the topic.

Despite the observed differences in CAM terminology and definitions, a systematic review about the evidence for the efficacy of CAM in the clinical context of RA [8] and the updated National Institute for Health and Care Excellence (NICE) guideline on management of RA [27], agree that CAM may provide short-term symptomatic benefit for the disease although there is little or no evidence for its long-term efficacy. Moreover, CAM treatments have the potential to interact with traditional drugs or to create treatments delays or withdrawals from more traditional treatments. These drawbacks can be minimized by treating patients within integrative settings with full transparency for all health care providers [7].

Finally, risk perception (RP) is defined as a multidimensional phenomenon that describes an individual's judgment of the likelihood of experiencing something unpleasant [28]. Health researchers have adopted the RP concept and have suggested that the average RP level for a

threat is related to the average level of the perceived characteristics of that particular threat, including its prevalence, controllability, preventability, and seriousness [29, 30]. The recognition of a significant risk to health in the face of the threat of complications can motivate patients to adopt preventive health behaviors [31]. Recent literature suggests that models of RP should separate deliberative RP, defined as systematic, logical and rule-based, from affective RP, which refer to the affect associated with the risk and is considered a critical component of judgments involving risks and uncertainty. Deliberative and affective RP should also be separated from experiential RP, which refer to rapid judgments made by integrating deliberative and affective information [32]. Published evidence from a meta-analysis reveals the relevance of these three conceptualizations of RP by illustrating that affective RP is related to preventive behaviors (in breast cancer) [33] and that interventions that successfully target these perception produce changes in behavior that may have an impact on health [34]. Moreover, research suggests that deliberative and affective components may interact and affect health behaviors, although the pattern of such interaction has been described as inconsistent [32]. RP has also been associated with unfavorable health behaviors due to judgment biases, such as unrealistic optimism or unrealistic pessimism, in which subjects underestimate or overestimate the likelihood of experiencing a negative event related to health [31, 35, 36].

There is limited RP-associated literature published in the field of rheumatic diseases. Two related qualitative studies in French patients with RA assessed the most frequent RA-related fears [37, 38]; however, fear and risk are not equivalent terms, though fear has been accepted as an associated dimension of RP [31, 39]. To date, there is no published literature that assesses the potential association between RP and CAM use. We recently developed and validated a RP questionnaire (RPQ) that was found to be valid, reliable, and feasible in evaluation of RP in our population of patients with RA [31].

Based on the above considerations, the primary aim of this study was to investigate if significant RP was associated with CAM use in Hispanic outpatients with RA and to describe additional associated factors. Secondary objectives were to describe patients' motivations for CAM use, satisfaction with CAM use, perceptions about CAM costs and safety, sources of information regarding CAM, and attitudes about CAM disclosure with primary rheumatologists.

## Material and methods

### Ethical considerations

The study was performed in compliance with the Declaration of Helsinki [40]. The Research Ethics Committee of the Instituto Nacional de Ciencias Médicas y Nutrición Salvador Zubirán (INCMyN-SZ) approved the study (Reference number: IRE-2901-19-20-1). All included patients participated in the informed consent process and provided written informed consent.

### Study design, setting, and study population

This cross-sectional study was performed between March and August 2019 at the outpatient clinic of the Department of Immunology and Rheumatology of INCMyN-SZ, a national referral center for rheumatic diseases in Mexico City.

Consecutive RA patients who were currently attending the outpatient clinic were invited to participate. A RA diagnosis was made based on the treating rheumatologists' criteria. Exclusion criteria included RA patients with Overlapping Syndrome (but secondary Sjögren's Syndrome), those receiving palliative care, and those with uncontrolled comorbid conditions, defined as having comorbid condition-related recent treatment modification or new treatment initiation.

Table 1. Summary of patient assessments and recorded information.

| Socio-demographic | RA-related | Treatment- related | Comorbid conditions | PROs |
|---|---|---|---|---|
| • Age, sex, years of scholarship, household type, labor information, urban vs. rural residence, health care access and socio-economic level. | • Rheumatoid factor status<br>• Disease duration<br>• Disease activity and remission status | • DMARD use and number of DMARDs/patient<br>• Corticosteroid use | • Comorbidities<br>• Charlson Comorbidity Index score | • Pain-VAS<br>• Overall disease-VAS<br>• Adherence-VAS<br>• RAPID-3<br>• RP<br>• CAM use |

RA = rheumatoid arthritis; DMARDs = disease-modifying anti-rheumatic drugs; PROs = patient-reported outcomes; VAS = visual analogue scale; RAPID-3 = Routine Assessment of Patient Index Data 3; RP = risk perception; CAM = complementary and alternative medicine.

## Patient assessments

All patients who consented to participate in the study were invited to a face-to-face interview to collect sociodemographic information; disease-, treatment-, and comorbid condition-related information; and patient-reported outcomes (PROs), including CAM use, as summarized in Table 1. When applicable, medical records were reviewed to corroborate the data provided by patients. All interviews were performed by three trained physicians who reviewed and agreed on CAM terminology before study initiation. Interviews were performed immediately after patients visited their primary rheumatologists, in a different location within the outpatient clinic that was suitable for clinical research. The primary rheumatologist was not present during interview and study assessments.

## Instrument descriptions

Standardized formats were used to retrieve all information. In addition, the following assessment tools were used. Instruments were scored by a single co-author with experience in PROs scoring and interpretation (ICY).

**Charlson score.** The Charlson Comorbidity Index was originally designed to measure the one-year mortality risk attributable to comorbidities in hospitalized patients. It is a weighted index that takes into account both the number and the seriousness of comorbid diseases. Higher scores indicate poorer one-year survival. All patients with RA scored at least one point [41].

**Pain-Visual Analogue Scale (VAS) and overall disease-VAS.** Both scales were used as recommended by the American College of Rheumatology (ACR) to evaluate pain and overall disease, respectively. The pain scale assessed "today" pain (instead of pain during a one-week period) on a 100 mm horizontal VAS, with "no pain" at the left end (corresponding to 0 mm), and "worst possible pain" at the right end (corresponding to 100 mm). Similarly, patient global/overall disease activity was also rated on a 0 to 100 mm horizontal VAS, with "worst possible disease activity" located at the right end of the scale and corresponding to 100 mm [42].

**Adherence-VAS.** A 0 to 100 mm scale was used to assess adherence, with 100 mm indicating the poorest adherence with RA-related overall treatment and located at the right end, meanwhile 0 mm was located at the left end that corresponded to the best adherence with RA-related overall treatment.

**Routine Assessment of Patients Index Score-3 (RAPID-3).** The RAPID-3 includes the following three measures: physical function, pain, and a patient global estimate evaluation. It has a raw score of 0 to 30 and an adjusted score of 0 to 10, with higher scores translating to higher disease activity. Four proposed severity categories, rather than disease activity categories, are also defined based on a 0 to 30 scale with cut-offs as follows: > 12 as high, 6.1 to 12.0 as moderate, 6.0 to 3.1 as low, and ≤ 3 as near-remission [43].

**Risk perception questionnaire, (S1 Appendix).** The RPQ is composed of 27 items distributed across the following five dimensions: likelihood to develop articular and extra-articular manifestations (nine items), likelihood to develop complications and/or comorbidities and disease severity (seven items), likelihood to develop unfavorable consequences (eight items), perception of personal responsibility to prevent and develop RA-related complications (two items), and perception of personal control over the disease (one item). It was constructed to integrate both patient and health care provider perspectives and has been found to be valid, reliable, and feasible for assessments of RP in our population. The RPQ score ranges from 0 to 100 mm, where 100 indicates the highest risk perception. Patients with a score ≥ 61.7 mm were considered to have significant RP [31].

**International Questionnaire on use of Alternative and Complementary Medicine (I-CAM-Q), (S2 Appendix).** A translated and cross-culturally adapted Argentinian version of the I-CAM-Q was used [44]. This Spanish version has a similar structure to the original version [45] with four main sections: (1) recent visits to different providers of CAM treatment; (2) CAM treatment received from a physician (MD); (3) consumption of medicinal products derived from herbs, vitamins, minerals, or homeopathic medicines; and (4) self-help practices implemented by patients. For the first section, patients were asked whether they had seen the providers within the past 12 months and, if so, were asked to indicate the number of times the health care providers were seen in the past 3 months. In section 2, patients were asked to indicate whether they had received any of five complementary treatments (manipulation, homeopathy, acupuncture, herbs, or spiritual healing) from a physician in the last 12 months and, if so, were asked to indicate the number of times in the past 3 months. In section 3, they were asked to list up to three products used within the past 12 months in each of four categories (herbal medicine, vitamins/minerals, homeopathy, or other supplements). They were also asked to identify the main reason for their last use of the products and to evaluate how helpful they found the products. Current use was also examined. Finally, in section 4 patients were asked about self-help practices (including praying for own health) using the same structure of questions from sections 1 and 2 (past 12 and 3 months, respectively). In addition, the frequency of use and level of benefit received for each item from each section were also explored [44, 45].

We provided an additional survey to patients to gather more detailed information related to CAM use, including the timing and motivations for CAM use in relation to RA diagnosis and institutional health care (seven items), CAM costs (three items), adverse events related to CAM use (one item), sources of information regarding CAM (one item), and patient perceptions of CAM disclosure to attendant rheumatologists (six items).

## Sample size calculation

In order to detect an effect size of 20% as an absolute difference in CAM use between patients with and without significant RP, we estimated the sample size using a two-tailed test with a 5% significance level and a power of 85%. The expected proportions were 40% and 20% in each group, respectively. The G*Power estimate was a total sample size of 220 patients. The CAM users versus non-users distribution that was obtained at study completion allowed us to have a power of 0.74 in a one-tailed test.

## Statistical analyses

We performed descriptive statistical analyses, presenting frequencies for categorical variables and measures of position and dispersion for numerical variables.

Significant RP was defined based on the 61.7-mm cut-off that corresponded to the 75th percentile from the data of patients included in the original description [31].

Patients CAM users were defined as patients who gave a positive answer to at least one of the four sections of the Spanish version of the I-CAM-Q; however, answers were restricted to the most recent period ("in the past of 3 months" for sections 1, 2, and 4, or "currently" for section 3). These time periods were selected in order to better represent current patient status regarding disease activity, pain, and disability. In addition, prayer was considered a form of CAM when patients prayed about their arthritis, as suggested in the original description of the survey development [4]. In Mexico, the most recent national survey found that 89.3% of citizens over 18 years of age declared themselves as Catholics [46]. As a result, the number of CAM users could potentially have been overestimated, since patients who profess a religion may not identify reasons for praying in the context of suffering from a chronic disease. Accordingly, we also calculated the frequency of CAM users excluding the prayer category and repeated analyses.

Patients CAM users were compared to non-CAM users. The Mann-Whitney U test was used to compare continuous variables without a normal distribution (Kolmogorov-Smirnov test). Fisher's exact test or the Chi-squared test were used to compare proportions.

Multiple logistic regression analysis was used to investigate factors associated with CAM use, which was considered the dependent variable. Variables included in the model were selected based on their statistical significance in the univariate analysis ($p \leq 0.10$) and their clinical relevance. The number of outcomes was also considered in order to avoid overfeeding the model. Previously, correlations between specific variables were analyzed and, when appropriate (Pearson correlation $\geq 0.80$), were selected according to clinical relevance. Significant RP was included in the model as a dichotomous variable (Yes/No). There were 0.1–3.5% of missing data identified and no imputation was performed.

All statistical tests were two-sided and evaluated at the 0.05 significance level. The statistical analyses were performed using the SPSS/PC program (v.21.0; Chicago, Illinois, USA).

# Results

## Population characteristics at study entry

During the study period, 266 patients were invited to participate in the study and 20 patients elected against participation due to lack of time for the interview. Characteristics of the 246 included patients are summarized in Table 2. Included patients were primarily middle-aged females with a median of 10.5 (IQR: 4–14) years of formal education. The median follow-up time was 16 (8–23) years. The majority of the patients in whom data were available (91.9%) had a positive rheumatoid factor (RF). Also, most patients had moderate disease activity based on the RAPID-3 score, while a minority were in remission (17.1%). The majority of the patients with available data (65.9%) had at least one additional comorbid condition and the median Charlson score was 1 (1–2). Almost all patients (94.3%) were receiving DMARDs and 25.5% of patients were on corticosteroids. Median (IQR) adherence-VAS was 81 mm (57.5–94), reflecting adequate adherence to RA-related treatment. The median pain-VAS and global-disease-VAS were 28 mm (7–60) and 30 mm (8–55), respectively, reflecting patients with substantial disease activity. Meanwhile, the median RPQ score was 48.2 (34.3–64.7) with 28.5% of patients having significant RP.

**Table 2. Population characteristics and their comparison in the subpopulations defined according to CAM users/non-users in the past 3 months.**

| | Study population, N = 246 | Patients with CAM use, N = 200 | Patients without CAM use, N = 46 | p |
|---|---|---|---|---|
| **Socio-demographic characteristics** | | | | |
| Age, years | 53.3 (45–63.3) | 52.5 (44.6–62.5) | 57.5 (48–65.3) | 0.181 |
| Females[1] | 222 (90.2) | 182 (91) | 40 (87) | 0.420 |
| Formal education, years | 10.5 (6–14) | 11 (6–14.8) | 9 (4.8–12) | 0.042 |
| Household work[1] | 144 (58.5) | 115 (57.5) | 29 (63) | 0.512 |
| Urban residence[1] | 213 (86.6) | 175 (87.5) | 38 (82.6) | 0.471 |
| Living together[1] | 146 (59.3) | 116 (58) | 30 (65.2) | 0.408 |
| Median-low socioeconomic level[1] | 226 (92.2) | 185 (92.5) | 41 (91.1) | 0.759 |
| Access to Federal health care system[1] | 53 (21.5) | 48 (24) | 5 (10.9) | 0.072 |
| **RA-related characteristics** | | | | |
| Disease duration, years | 16 (8–23) | 17 (8–23) | 12.5 (7.8–23.5) | 0.591 |
| RF+[1] (160 data available) | 147 (91.9) | 116 (92.1) | 31 (91.2) | 1 |
| RAPID-3 score (245 data available) | 10.7 (5–16) | 11.3 (6–16.7) | 7.5 (2.9–13.5) | 0.030 |
| Remission status (RAPID-3 ≤3) (245 data available)[1] | 42 (17.1) | 29 (14.6) | 13 (28.3) | 0.048 |
| **Comorbidity** | | | | |
| Comorbid conditions[1] | 162 (65.9) | 127 (63.5) | 35 (76.1) | 0.122 |
| Charlson score (245 data available) | 1 (1–2) | 1 (1–1) | 1 (1–2) | 0.091 |
| **RA-related treatment** | | | | |
| DMARDs use[1] | 232 (94.3) | 187 (93.5) | 45 (97.8) | 0.478 |
| Corticosteroids use[1] | 62 (25.5) | 55 (27.9) | 7 (15.2) | 0.091 |
| N° of DMARDs/patient | 2 (1–2) | 2 (1–2) | 1 (1–2) | 0.252 |
| **PROs** | | | | |
| Pain-VAS (238 data available) | 28 (7–60) | 33 (8–61.5) | 17.5 (5–52.2) | 0.071 |
| Global-disease-VAS (238 data available) | 30 (8–55) | 32.5 (9–56.8) | 17 (4.8–47.3) | 0.048 |
| Adherence-VAS (245 data available) | 81 (57.5–94) | 80 (58–94) | 82.5 (55.3–97) | 0.516 |
| **Risk perception** | | | | |
| RPQ score | 48.2 (34.3–64.7) | 51.6 (38.2–65.3) | 36.9 (26.3–49.3) | ≤0.0001 |
| Significant RP[1] | 70 (28.5) | 62 (31) | 8 (17.4) | 0.072 |

Data presented as median (IQR) unless otherwise indicated.

[1]Number (%) of patients. RF = rheumatoid factor. RAPID-3 = Routine Assessment of Patients Index Score-3. DMARD = disease modifying anti-rheumatic drugs. PROs = patient-reported outcomes. VAS = visual analogue scale. RPQ score = risk perception questionnaire score.

### Primary objective: The impact of significant RP on CAM use and additional associated factors

According to the previous definition, there were 200 patients who were determined to be CAM users (81.3%), although this number was reduced to 92 (37.4%) when the prayer category was excluded.

Table 2 summarizes results from comparisons of the selected characteristics between CAM users and non-users. In summary, patients in the former group had more years of formal education, higher RAPID-3 scores, were less frequently in remission status, and rated higher on the global-disease-VAS and the RPQ score. They also tended to have more access to the federal health care system, lower Charlson scores, more frequent use of corticosteroids, higher scores on the pain-VAS, and had more frequently significant RP.

The following variables were included in the multiple logistic regression analysis to identify factors associated with CAM use: years of formal education, access to federal health care

**Table 3. Comparison of CAM users and non-users when prayer was excluded from CAM categories.**

| | Patients with CAM use, N = 92 | Patients without CAM use, N = 154 | p |
|---|---|---|---|
| **Socio-demographic characteristics** | | | |
| Age, years | 51 (42.4–58.6) | 56 (46.7–65.2) | 0.005 |
| Females[1] | 88 (95.7) | 134 (87) | 0.028 |
| Formal education, years | 12 (9–16) | 9 (6–12) | ≤0.001 |
| Household work[1] | 49 (53.3) | 95 (61.7) | 0.229 |
| Urban residence[1] | 84 (91.3) | 129 (83.8) | 0.122 |
| Living together[1] | 54 (58.7) | 92 (59.7) | 0.894 |
| Median-low socioeconomic level[1] | 81 (88) | 145 (94.8) | 0.082 |
| Access to Federal health care system[1] | 29 (31.5) | 24 (15.6) | 0.004 |
| **RA-related characteristics** | | | |
| Disease duration, years | 15 (8–21.8) | 17 (8–23.3) | 0.356 |
| RF+[1] (160 data available) | 46 (93.9) | 101 (91) | 0.756 |
| RAPID-3 score (245 data available) | 10.4 (5.2–17.2) | 10.7 (4.9–16) | 0.924 |
| Remission status (RAPID-3 ≤3) (245 data available)[1] | 16 (17.4) | 26 (17) | 1 |
| **Comorbidity** | | | |
| Comorbid conditions[1] | 50 (54.3) | 112 (72.7) | 0.004 |
| Charlson score (245 data available) | 1 (1–1) | 1 (1–2) | 0.263 |
| **RA-related treatment** | | | |
| DMARDs use[1] | 83 (90.2) | 149 (96.8) | 0.045 |
| Corticosteroids use[1] | 24 (27) | 38 (24.7) | 0.760 |
| N° of DMARDs/patient | 2 (1–2) | 2 (1–2) | 0.800 |
| **PROs** | | | |
| Pain-VAS (238 data available) | 32 (10–60) | 26 (6–59) | 0.466 |
| Global-disease-VAS (238 data available) | 30 (8–65) | 27 (8–52) | 0.562 |
| Adherence-VAS (245 data available) | 80 (55–94) | 82 (59.3–95) | 0.418 |
| **Risk perception** | | | |
| RPQ score | 53.1 (38.8–68) | 46.6 (32.7–61.5) | 0.040 |
| Significant RP[1] | 33 (35.9) | 37 (24) | 0.058 |

Data presented as median (IQR) unless otherwise indicated.

[1]Number (%) of patients. RF = rheumatoid factor. RAPID-3 = Routine Assessment of Patients Index Score-3. DMARD = disease modifying anti-rheumatic drugs.

PROs = patient-reported outcomes. VAS = visual analogue scale. RPQ score = risk perception questionnaire score.

system, remission status defined by RAPID-3 (highly correlated to RAPID-3 score and to both patient VAS scores), Charlson score, corticosteroid use, and significant RP (highly correlated to RPQ score); in addition, female sex and adherence-VAS score were forced into the analysis. Significant RP (OR: 2.388, 95% CI: 1.044–5.464, $p$ = 0.039) and access to the federal health care system (OR: 2.916, 95% CI: 1.081–7.866, $p$ = 0.035) were the only variables significantly associated with CAM use in the past 3 months.

Finally, Table 3 summarizes comparisons of selected variables between CAM users and non-users when prayer was excluded from CAM categories. Patients in the former group were younger, more frequently female, and had more years of formal education, more access to federal health care, less comorbid conditions, higher RPQ scores, and had more frequently significant RP. Also, these patients tended to more frequently have urban residence and to receive DMARDs less frequently.

The following variables were included in the multiple logistic regression analysis to identify factors associated with CAM use when the prayer category was excluded: age, female sex, years

**Table 4. Multiple logistic regression analysis to predict CAM use (prayer category excluded).**

|  | OR | 95% CI | p value |
|---|---|---|---|
| Significant RP | 2.222 | 1.208–4.087 | 0.010 |
| Years of formal education | 1.095 | 1.031–1.163 | 0.003 |
| Access to Federal health care system | 2.388 | 1.208–4.718 | 0.012 |
| Female sex | 4.023 | 1.245–12.994 | 0.020 |

RP = risk perception. OR = Odds ratio. CI = confidence interval.

of formal education, access to federal health care system, comorbid conditions, DMARD use, and significant RP (highly correlated to RPQ score); in addition, adherence-VAS score was forced into the analysis. Table 4 shows that CAM use was associated with significant RP, female sex, years of formal education, and access to the federal health care system.

## Secondary objectives

The complementary (to the Spanish for Argentinian version of the I-CAM-Q) survey was given to the 92 CAM users (prayer excluded from CAM categories). Data from 86 patients (93.4%) were available.

**Timing and motivation for CAM use.** All the patients interviewed combined CAM use with institutional health care; twenty-two (25.6%) were already CAM users before their RA diagnoses, while the remaining patients became regular CAM users after their RA diagnoses. Among these patients, few (22.1%) used CAM only when the disease was uncontrolled. Finally, the majority of patients (70 [81.4%]) reported that CAM use was primarily related to their RA diagnoses.

**Patient perceptions about CAM costs and safety.** The majority of patients perceived CAM treatment to be cheaper than "traditional RA-related treatment" (49 [57%]) and few (17.4% and 18.6%) perceived the cost to be similar or more expensive, respectively. Also, most of the patients (57 [66.3%]) felt CAM-related costs were within the category of "reasonable-cheap." The mean expenditures for CAM remedies and provider fees were equivalent to 2 days (0.56–5.84) of the official minimum daily wage per patient. Finally, 20 patients (23.3%) reported nonadherence to CAM treatment.

Only three patients (3.5%) reported adverse events related to CAM use and all adverse events spontaneously resolved.

**Patient sources of information regarding CAM.** The majority of patients (68 [79.1%]) reported that friends and family were the primary sources of information/recommendation for CAM use, while few (5 [5.8%]) received the recommendation from other RA patients.

**Patient attitudes about CAM disclosure to their primary rheumatologist.** The majority of patients (63 [73.3%]) agreed that CAM use should be disclosed to the primary rheumatologist; however, few (21 [31.4%]) actually disclosed it, despite sharing information about CAM use with their friends and family (76.7% of the users). The main motivations for nondisclosure to the primary rheumatologist were fear of being reproached (41.9%) and fear of having institutional health care suspended (19.8%). Meanwhile, almost all patients (81 [94.2%]) referred to being satisfied with the care provided by their rheumatologist. Finally, CAM users who disclosed to rheumatologists (N = 27) were compared to those who did not (N = 59). The former group had significantly more years of formal education than their counterparts (15 years [12–17] versus 11 [8–14], p = 0.003).

**CAM modalities and patient perceived benefits.** Fig 1 summarizes most significant results. There were 29 patients (14.5%) who reported recent visits to CAM providers, most

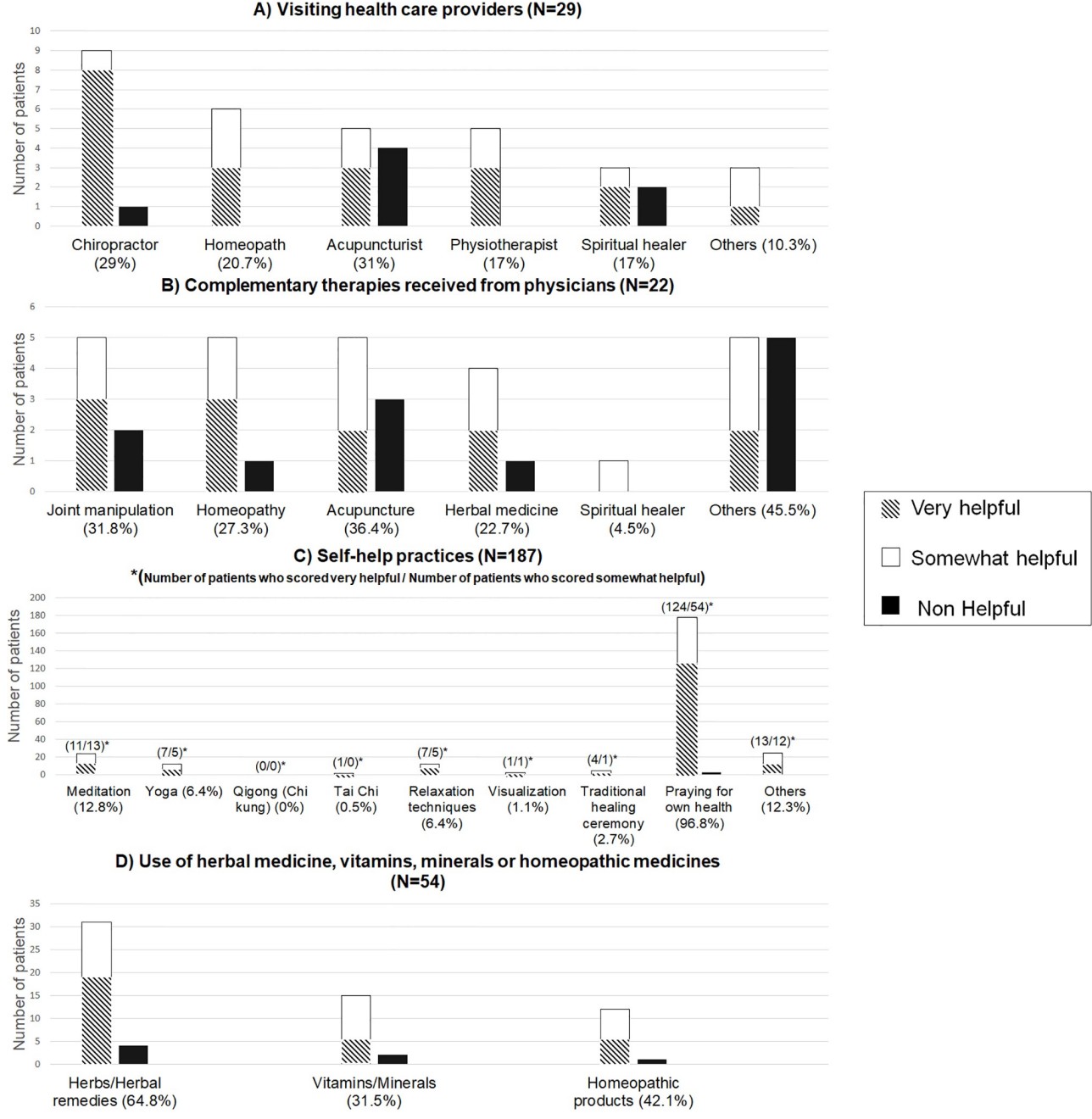

**Fig 1. CAM modalities and patient´s perceived benefits.**

frequently to an acupuncturist (31%) or a chiropractor (29%). There were 22 patients (11%) who reported that they were referred for CAM treatment by a physician; the most frequent referral modalities were acupuncture (36.4%) and joint manipulation (31.8%). Fifty-four patients (27%) reported the use of herbal medicines, vitamins/minerals, or homeopathic medicines, with herbal remedies most frequently reported (in 64.8% of the patients) and then vitamins/minerals (in 31.5%). Finally, almost all patients (187 [93.5%]) reported self-help

practices, with praying for one's own health most frequently reported (96.8%), followed by meditation (12.8%), yoga (6.4%), and relaxation techniques (6.4%).

Patients perceived CAM use as very/somewhat helpful to a variable degree, from 55.5% to 100%. Self-help practices were almost universally perceived as very/somewhat helpful (95.7–100% of the modalities), as were herbal remedies, vitamin/minerals, and homeopathic remedies (88.2–92.3%). Meanwhile, acupuncture was perceived as very/somewhat helpful in only 62.5% of patients and visiting an acupuncturist in 55.5% of patients.

## Discussion

The present study revealed that significant RP, in addition to access to a federal health care system, were significantly associated with CAM use in Hispanic outpatients with RA at a tertiary care center. Similar results were obtained when prayer was omitted from the CAM definition, although additional associated factors were also identified, including female sex and years of formal education. This study also revealed that the majority of patients combined CAM use with institutional healthcare and that patients most frequently became CAM users after RA was diagnosed. Patients perceived CAM-related costs to be cheaper than traditional medicine and they considered CAM modalities to usually be safe. Their primary sources of CAM-related information were friends and family. Also, a minority of CAM users disclosed their use with their primary rheumatologist and their main motivation for nondisclosure was fear of unfavorable consequences. Finally, patients used different CAM modalities and remedies, although there were variations in the perceived benefits of the different modalities.

The formation of RP relies on the ability to produce, understand, and use numerical information; however, a number of additional factors also contribute, including patients' personal experiences, salience of available and close examples, and affective factors [32]. In addition, RP may be influenced by contextual factors, and tend to be higher (or more pessimistic) when a health threat is perceived to be uncontrollable or dreaded [47]. This theoretical RP framework may support the association found in our patients between significant (and pessimistic) RP and CAM use, as patients with significant RP may have perceived their disease as more threatening when compared to their counterparts and may, therefore, have decided to add CAM interventions to traditional medicine. In previous studies in the oncology field, patients perceived CAM use as the opportunity to be more active with their treatment and care, to feel they were gaining control over their illness, and to improve their chances of benefit from conventional medicines [48]. In addition, patients experienced a sense of urgency as a result of their illness; however, use of the publicly funded health care system was restricted by waiting lists and limited time for consultations, a context that did not apply to CAM interventions [48]. All of these considerations may be relevant for patients with chronic, painful, and potentially disabling diseases such as RA, particularly for those who rated themselves with significant RP, and may interact to favor CAM use. In fact, a qualitative study performed in French patients with RA (and spondylarthritis) highlighted that RA patients held a core set of beliefs and apprehensions/fears, many of which were unappropriated from a medical point of view, which may be considered a surrogate of unrealistic and pessimistic RP; as already mentioned, although not equivalent to risk, fear contributes to RP construct. Finally, our results may be particularly relevant and unique to our patients, as nationality influences RA patients' perceptions about physician trustworthiness and the choice of the RA priority domains [38].

Our study also identified additional factors that were significantly associated with CAM use, although factors differed depending on whether the prayer category was excluded (or not) from the CAM use definition. This is not surprising considering the currently recognized importance of spirituality and existential concerns in health care settings, beyond its initially

limited applications with terminally ill and older patients [49]. Spirituality, religiousness and existential concerns have become a major component of health-related quality of life and thus are part of patient-reported-outcomes-measures [50]; spiritual orientation can help people to cope with the consequences of a serious disease [51], meanwhile the value of elements such as faith, hope and compassion in the healing process is increasingly recognized by patients and physicians [52]. Finally, spirituality might have identified patients with unique characteristics and shaped the results from regression models.

Access to the federal health care system was consistently associated to CAM use meanwhile female sex (with the highest risk), and years of formal education appeared relevant when the prayer was omitted from the CAM use definition. Previous studies have shown that women [13, 15, 17, 21, 53] with higher education levels [10, 11, 13] fit the profile of users of CAM, either as an exclusive treatment or concomitantly with conventional medicine; however, no significant gender differences in arthritis-focused CAM consumption have also been described [21]. Moreover, lower education levels have also been associated with higher CAM use [54]. In Mexico, Ramos-Remus et al. [14] found that the use of alternative therapies was associated with lower education levels and slightly higher disability scores in 300 consecutive patients with rheumatic diseases, including 122 patients with RA. Álvarez-Hernández et al. [6] found an association with longer disease duration in 800 consecutive patients attending an outpatient rheumatology clinic for the first time, among whom 22.3% had RA. Both studies were performed almost 15 years ago in particular subpopulations of younger and slightly less educated patients than in our study, which may explain discrepancies in our results. In addition, in the former studies, CAM use was assessed based on a face-to-face interview [14] or a self-administered questionnaire [6]. It needs to be highlighted that the association found between CAM use and access to a federal health care system has not been previously described and could be indicative of patients' desires to seek continuous traditional health care, while bringing their cultural beliefs into their health seeking behavior [22]. This conclusion is supported by the findings that the majority of the patients became CAM users after RA was diagnosed (74.4%), used CAM primarily due to a RA diagnosis (81.4%), and did not limit CAM use to control of disease flares (87.9%). Of note, the study by Ramos-Remus et al. [14] was performed in patients who all had access to a federal health care system, and similar results regarding patients' motivations for CAM use were identified. Finally, a study performed in 480 elderly patients with arthritis and additional comorbid conditions showed an association between the use of CAM for arthritis and a higher use of traditional health care resources [55], which is conceptually related to general access to health care resources.

Our patients perceived that the costs associated with CAM use were less than traditional DMARDs and, of note, patients who attended our institution had to pay for their own medications. The literature highlights that costs related to CAM use varies between countries [21], possibly because some countries provide more access to CAM modalities within their overall health care systems [56]. Interestingly, one study found that in patients with fibromyalgia, high CAM-related costs were a relevant reason for not using these treatment strategies [57]. Ramos-Remus et al. [14] noted that costs of provider services and alternative remedies were high considering the low incomes of their patients, although patient perceptions were not assessed.

Our RA patients perceived CAM strategies as safe. Complementary therapies have reportedly been perceived by physicians [20] and patients as having few, if any, side effects [34, 57], despite evidence that CAM treatment can cause gastrointestinal side effects [58]. Additional serious adverse effects have also been described, and have provoked warnings by regulatory agencies against their use, particularly against *Tripterygium wilfordii* (thunder god wine) [16]. Moreover, patients who concomitantly use conventional and CAM medicines may

misattribute side effects to conventional DMARDs and stop their usage [16]. In the Mexican study from Ramos-Remus et al., side effects associated with the use of alternative treatments were reported in up to 16% of their patients, although all the patients recovered without specific treatments [14].

The most important sources of information guiding our patients to use CAM were friends and family members, as has been previously described [16, 21, 59–61]. Cultural and personal backgrounds have been reported to be influential factors for CAM use and patients may receive recommendations to use CAM by other family members who have had previous experiences with CAM [15]. Patients primarily rely on their social networks for information regarding CAM strategies [62, 63] and are willing to try therapies even without the approval of their physicians [64]. In our study, while the majority of CAM users referred to being satisfied with the care provided by their rheumatologists (94.2%) and agreed that CAM should be disclosed to him/her (73.3%), few (31.4%) patients actually disclosed CAM use. Fear of negative consequences was the main reason in up to 60% of our patients. The lack of disclosure about CAM use by patients is almost universal [22], with similar rates reported in some studies [12, 14, 65] and lower rates described in others [16]. A negative response regarding CAM from the medical practitioner has previously been identified as a factor associated with nondisclosure, though rarely reported in other studies [22, 23, 60]. This discrepancy with our study may be related to a greater acceptance of alternative therapies in some countries, as has been described in Europe [61]. Also, study patients who disclosed CAM use to their rheumatologists had more years of formal education, a finding that has been highlighted in a recent review [21] and reproduced in patients with osteoarthritis [19].

Finally, our identified rates and modalities of CAM use were within the range of previous studies, including local descriptions [6, 12, 21, 22, 55, 66, 67]. Much higher levels of CAM use were found when prayer was included, as previously reported [19]. Moreover, most patients perceived the therapies to be beneficial, in accordance with previous reviews [21, 56]. On the other hand, a more conservative figure has been reported when physicians assessed the benefits of CAM [8, 9, 12, 18, 20, 56], despite evidence that rheumatologists have a widespread favorable opinion toward many, but not all, types of CAM [68].

This study had some limitations to be addressed. First, the study had a cross-sectional design and therefore only associations can be inferred; accordingly, a temporal or causal association between significant RP and CAM use is debatable. Second, we focused on significant RP as a potential associated factor with CAM use, and a limited number of additional factors were examined. Depression has previously been associated with CAM use [15], as has health literacy [69]; however, their presence was not assessed in this study. Third, Mexican RA patients may have some cultural familiarity with specific CAM strategies and, therefore, may not recognize them as CAM. Fourth, we assessed factors associated with CAM use as a whole category; however, there is evidence that associated factors may differ by the different categories of CAM use [67, 69]. Fifth, the study was conducted in a single academic center, where patients may have had higher levels of comorbid conditions that may increase CAM use. Finally, CAM use was assessed through the application of the Spanish for Argentina translated and adapted version of the I-CAM-Q and a formal validation of this questionnaire is lacking; in addition, interview and instruments application were done by 3 trained physicians in whom repeatability (intra-observer) and reproducibility (inter-observer) were not examined.

## Conclusions

In the present study, we showed that significant RP was consistently associated with recent CAM use in Hispanic outpatients with RA. Our study adds relevant and practical information

to the existing knowledge base about how an RA patient´s perception of the disease may significantly influence his or her self-care behavior, and emphasizes that rheumatologists should clearly communicate with their patients regarding CAM use. Our single-center study is limited by the sample size and needs to be replicated in a larger sample.

## Supporting information

**S1 Appendix. Risk Perception Questionnaire (RPQ).**
(PDF)

**S2 Appendix. International Questionnaire on use of Alternative and Complementary Medicine (I-CAM-Q).**
(PDF)

## Author Contributions

**Conceptualization:** Irazú Contreras-Yáñez, Andrea Robledo-Torres, Claudia Cáceres-Giles, Salvador Valverde-Hernández, Diana Padilla-Ortiz, Guillermo Arturo Guaracha-Basáñez, Virginia Pascual-Ramos.

**Data curation:** Irazú Contreras-Yáñez.

**Formal analysis:** Irazú Contreras-Yáñez, Virginia Pascual-Ramos.

**Investigation:** Irazú Contreras-Yáñez, Ángel Cabrera-Vanegas, Andrea Robledo-Torres, Claudia Cáceres-Giles, Salvador Valverde-Hernández, Diana Padilla-Ortiz, Guillermo Arturo Guaracha-Basáñez.

**Methodology:** Irazú Contreras-Yáñez, Ángel Cabrera-Vanegas, Andrea Robledo-Torres, Claudia Cáceres-Giles, Salvador Valverde-Hernández, Diana Padilla-Ortiz, Guillermo Arturo Guaracha-Basáñez, Virginia Pascual-Ramos.

**Supervision:** Irazú Contreras-Yáñez, Virginia Pascual-Ramos.

**Visualization:** Irazú Contreras-Yáñez, Virginia Pascual-Ramos.

**Writing – original draft:** Irazú Contreras-Yáñez, Virginia Pascual-Ramos.

**Writing – review & editing:** Irazú Contreras-Yáñez, Ángel Cabrera-Vanegas, Andrea Robledo-Torres, Claudia Cáceres-Giles, Salvador Valverde-Hernández, Diana Padilla-Ortiz, Guillermo Arturo Guaracha-Basáñez, Virginia Pascual-Ramos.

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
