## [Decision Letter · Decision Letter 0]

12 Jun 2020

PONE-D-20-01920

Association of risk perception with the use of complementary and alternative medicine: a cross-sectional study in Hispanic patients with rheumatoid arthritis

PLOS ONE

Dear Dr Pascual-Ramos,

Thank you for submitting your manuscript to PLOS ONE. After careful consideration, we feel that it has merit but does not fully meet PLOS ONE’s publication criteria as it currently stands. Therefore, we invite you to submit a revised version of the manuscript that addresses the points raised during the review process.

We would appreciate receiving your revised manuscript by Jul 02 2020 11:59PM. To enhance the reproducibility of your results, we recommend that if applicable you deposit your laboratory protocols in protocols.io, where a protocol can be assigned its own identifier (DOI) such that it can be cited independently in the future. For instructions see: http://journals.plos.org/plosone/s/submission-guidelines#loc-laboratory-protocols

We look forward to receiving your revised manuscript.

Kind regards,

Daniel Steffens, PhD

Academic Editor

PLOS ONE

2. Please include additional information regarding the survey or questionnaire used in the study and ensure that you have provided sufficient details that others could replicate the analyses. If you developed and/or translated a questionnaire as part of this study and it is not under a copyright license more restrictive than Creative Commons Attribution (CC-BY), please include a copy, in both the original language and English, as Supporting Information.

Reviewers' comments:

Reviewer's Responses to Questions

**Comments to the Author**

1. Is the manuscript technically sound, and do the data support the conclusions?

Reviewer #1: Yes

Reviewer #2: Partly

Reviewer #3: Partly

2. Has the statistical analysis been performed appropriately and rigorously? 

Reviewer #1: Yes

Reviewer #2: Yes

Reviewer #3: Yes

3. Have the authors made all data underlying the findings in their manuscript fully available?

Reviewer #1: Yes

Reviewer #2: Yes

Reviewer #3: Yes

4. Is the manuscript presented in an intelligible fashion and written in standard English?

Reviewer #1: Yes

Reviewer #2: Yes

Reviewer #3: Yes

5. Review Comments to the Author

Reviewer #1: (22-23) I'd prefer a better link between those two sentences, just personal opinion;

(62) "and destruction of the joints" redundant; (62) "challenges" I'd prefer 'impairment';

(84) maybe go to new paragraph after the full stop;

(333) "nagged" use a better term;

(335-336) is that data collected in anonymous form?

(425) maybe is better 'to seek continuous medical assessment/reassurations?'

Overall it's good in my opinion.

Reviewer #2: This manuscript presents a cross-sectional study investigating the association of risk perception (RP) among rheumatoid arthritis (RA) patients using complementary and alternative medicine (CAM) in a Hispanic RA outpatient population. Through interviewing a consecutive sample of patients at this single centre, they attempted to evaluate risk perception along with other factors which they believed to be associated with influencing CAM use. The authors identified limited RP-associated literature pertaining to rheumatic disease and therefore a gap in the evidence to address. The conclusion of this study appears to indicate that RP was associated to recent (within 3 month) CAM use in Hispanic RA outpatients. The strengths of this study were a clear aim, following a well-documented & reproducible methodology and appropriate sub-analyses given the included population.

As above, the authors have presented a reproducible study that investigated an area of interest to the profession. Overall this was well put together and this reviewer would recommend the editor to accept this submission pending revision pertaining to caveats outlined in attached document.

Reviewer #3: Introduction

Paragraph 2, lines 84-86

What references support the statement of short-term benefits? Please insert the references that support this information.

Care must be taken when asserting short-term effectiveness using the reference provided. It reported paucity of randomized controlled trials in the area. The quality of the evidence does not allow to affirm the presence/absence of effectiveness of the intervention.

Methods

Adherence-VAS, line 163

Were the evaluating physicians present in the treatment of patients?

If so, wouldn't that be a factor that would induce the patient's response?

Please indicate if the evaluating physicians were the same responsible for the care of the participants. If so, present it as a limitation of the study.

Patient assessments, line 120

Did the evaluators show inter- and intra-evaluator reliability for the application of the evaluation instruments? Training alone does not guarantee the evaluator's reliability.

Please provide the evaluator's reliability data.

If reliability does not exist, present it as a limitation of the study.

Results

Paragraph 1, line 247

266 participants were registering. However, the sample size was estimated at 220.

Please explain the reason for registering extra participants.

Primary objective: the impact of RP on CAM use and additional associated factors

Paragraph 2, lines 84-86

The authors state that participants in the first group "ended to have more access to the federal health care system, lower Charlson scores, more frequent use of corticosteroids, higher scores on the pain-VAS, and more significant RP". However, even though the absolute numbers were different, there was no statistical difference between the groups. In this sense, there is statistical equality between the two groups.

Please consider this information

Paragraph 4, line 288

The authors claim that the participants in the first group had higher scores for more significant PR. However, even though the absolute numbers were different, there was no statistical difference between the groups. Consider the previous comment.

Discussion

General

Authors can go more directly to the focus. Concepts such as PR can be better addressed in the introduction. In this way, there will be more space for important discussions such as: difference in results when prayer was excluded from CAM; Mechanisms by which PR can influence CAM use.

Paragraph 2

I believe that this paragraph should have a main focus on the possible mechanism by which PR influences CAM. I believe that conceptual terms of PR can be better addressed in the introduction.

Paragraph 6

Use a reference to support the statement:

“Patients primarily rely on their social networks for information regarding CAM strategies and are willing to try therapies even without the approval of their physicians.”

Conclusion

The conclusion can go more directly to the study's findings. Previous study information is covered in the discussion.

6. PLOS authors have the option to publish the peer review history of their article (what does this mean?). If published, this will include your full peer review and any attached files.

Reviewer #1: Yes: Stefano Di Donato

Reviewer #2: No

Reviewer #3: No

---

## [Author Response · Author response to Decision Letter 0]

25 Jun 2020

Association of risk perception with the use of complementary and alternative medicine: a cross-sectional study in Hispanic patients with rheumatoid arthritis” (PONE-D-20-01920)

Authors’ responses 

“Association of risk perception with the use of complementary and alternative medicine: a cross-sectional study in Hispanic patients with rheumatoid arthritis (PONE-D-20-01920)”.

Response: We have reviewed PLOS ONE requirements. 

2. Please include additional information regarding the survey or questionnaire used in the study and ensure that you have provided sufficient details that others could replicate the analyses. If you developed and/or translated a questionnaire as part of this study and it is not under a copyright license more restrictive than Creative Commons Attribution (CC-BY), please include a copy, in both the original language and English, as Supporting Information.

Response: We have added 2 files as Supporting information (S1 and S2 Appendix), with the RPQ in both languages, Spanish and English, in a table format and with the Spanish I-CAM-Q. (Lines 199, 220, 718-721).

Reviewer #1: 

We appreciate the reviewer comments. 

1. (22-23) I'd prefer a better link between those two sentences, just personal opinion.

Response. We propose an updated paragraph (Lines 22-25).

(62) "and destruction of the joints" redundant; (62) "challenges" I'd prefer 'impairment'.

Response. We propose the following sentence: ”Rheumatoid arthritis (RA) is a systemic inflammatory disorder with articular and extra-articular involvement that, if not properly controlled, can lead to significant structural damage, functional impairment, disability, reduced quality of life, and increased mortality [1-3]” (Lines 61-62).

2. (84) maybe go to new paragraph after the full stop (Line 87).

Response: We have adopted the suggestion. 

3. (333) "nagged" use a better term;

Response: We propose the term “reproached” (Line 354).

4. (335-336”) is that data collected in anonymous form?

Response: Data were not anonymous, but the physician in charge of collecting the information was not the primary rheumatologist in charge of patients’ health care delivery.

6. (425) maybe is better 'to seek continuous medical assessment / reassurations?'

Response: We propose the following sentence “It needs to be highlighted that the association found between CAM use and access to a federal health care system has not been previously described and could be indicative of patients’ desires to seek continuous traditional health care, while bringing their cultural beliefs into their health seeking behavior [22] “ (Lines 442-446).

Overall it's good in my opinion.

Reviewer #2: 

This manuscript presents a cross-sectional study investigating the association of risk perception (RP) among rheumatoid arthritis (RA) patients using complementary and alternative medicine (CAM) in a Hispanic RA outpatient population. Through interviewing a consecutive sample of patients at this single centre, they attempted to evaluate risk perception along with other factors which they believed to be associated with influencing CAM use. The authors identified limited RP-associated literature pertaining to rheumatic disease and therefore a gap in the evidence to address. The conclusion of this study appears to indicate that RP was associated to recent (within 3 month) CAM use in Hispanic RA outpatients. The strengths of this study were a clear aim, following a well-documented & reproducible methodology and appropriate sub-analyses given the included population.

As above, the authors have presented a reproducible study that investigated an area of interest to the profession. Overall this was well put together and this reviewer would recommend the editor to accept this submission pending revision pertaining to caveats outlined in attached document.

Major revisions

• Abstract results refer to sample with prayer included as CAM. Prayer is not usually referred to as CAM, though the authors do make assertions that their population may be influenced by it. Suggest focus of the paper should be related to the prayer excluded sample. This is further supported by the I-CAM-Q not having any reference to prayer and also only being requested from patients who were in the prayer excluded CAM group in this study.

Response: We disagree with the reviewer. The ICAM-Q section 4 “Self-help practices” includes prayer for own health (see reference 45). We have added the following sentence to make this point more clear and to defend that both analysis were convenient: “In addition, prayer was considered a form of CAM when patients prayed about their arthritis, as suggested in the original description of the survey development [45]” (Lines 242-244). 

To have an overall estimation of CAM use in our population, prayer (for own health) was considered; this strategy allowed us to compare our results with those from other populations. Nonetheless, considering that Mexican referred themselves primary as Catholics, and the fact that it may be difficult to differentiate “praying” from “praying for own health”, we decided to exclude the prayer category, to assess main objectives. 

• While the authors conclusion does seem to reflect the results they found (minus that access to a federal health care system was found to be associated on both analyses as on Line 359; Discussion), as per their sample size calculation section (Line 214-215), this sample is powered to 0.74 in a one-tailed test. Therefore the conclusion should be amended to provide more uncertainty regarding these associations and suggest further investigation is necessary with a larger comparative sample (non-CAM user). 

Response: We have updated the conclusion according to the reviewer suggestion (Lines 519-520).

• Given the sample size calculation section (Line 209-215), it would be helpful to provide an explanation as to why the full sample was not collected (assuming an equal distribution of the calculated 220 patient, i.e. 110 in each group).

Response: The full sample size (calculated) was collected; nonetheless, our estimate of CAM user distribution between patients with and without RP was different from that obtained; we did not assume equal distribution of CAM use in the patients with and without RP; we did consider 40% vs. 20% of CAM users in the groups defined according to significant RP. At study completion, the final distribution of CAM users in either group differed from our estimate (we obtained 89% vs. 78%) which allowed us the power described (0.74, slightly below the recommended 0.80). 

• Table 1 (Line 143) may not be set out in line with the usual journal criteria. A mix between sentence case and bullet points makes it difficult to follow the contents of this table. The authors have also chosen to use “*” as their bullet point which is usually associated with footnotes to tables. Suggest change each data point to a bullet point with an actual bullet instead of “*”.

Response: We have updated table 1 format.

• The Figure provided was extremely blurry and hard to interpret. A cleared image would be required to properly evaluate the information. Still, the authors have chosen to collapse if patients believed CAM was “very helpful” or “somewhat helpful” in the graph. Given these are very different responses, it may be helpful to have these represented as separate categories. It seems fine to collapse them in text (Line 353-357). 

• As per above, it might be the quality of the Figure, but it seems in the "complementary therapies received from physicians” graph, the joint manipulation bar is thinner that everything else. This reviewer is unclear if there is missing/cut off information or if this is just a sizing error.

Response: We propose a new figure according to reviewer suggestions. 

Minor revisions

• Acronyms should not start sentences (e.g. Line 34).

Response: We have adopted the suggestion.

• On Line 76-77, the authors provide the WHO definition of CAM as “a broad set of health care practices that are not part of the country´s own tradition and are not integrated into the dominant healthcare system”. The definition appears to refer to CAM being defined based on country-specific views of mainstream medicine. However on Line 81-84, the authors say 3 certain CAM approaches are ingrained in certain countries and then say that these methods “may not be appropriately classified under the WHO definition”. This reviewer is unclear on the message the authors are trying to convey.

Response: We pretend to highlight the lack of uniformity in CAM definition and CAM terminology that prevents consistent data analysis and comparison between published studies. We have updated the paragraph to better address this message (Lines 72-86). 

• The explanation of the Pain-Visual Analogue Scale (VAS) and overall disease-VAS (Line 155-162) was extremely confusing in terms of which end (left vs. right) was equal to 0 or 100. It is understandable that the authors used a referenced method, but it would be helpful if the reader could be assured that the scoring clinician was not confused by this. Possible re-iteration that the clinician scoring this was trained it how to do this on Line 134-135.

Response: We have updated the paragraph as follows: Both scales were used as recommended by the American College of Rheumatology (ACR) to evaluate pain and overall disease, respectively. The pain scale assessed “today” pain (instead of pain during a one-week period) on a 100 mm horizontal VAS, with “no pain” at the left end (corresponding to 0 mm), and “worst possible pain” at the right end (corresponding to 100 mm). Similarly, patient global/overall disease activity was also rated on a 0 to 100 mm horizontal VAS, with “worst possible disease activity” located at the right end of the scale and corresponding to 100 mm [42](Lines 170-176). 

In addition we have added the following sentence: “Instruments were scored by a single co-author with experience in PROs scoring and interpretation (ICY)” (Lines 162-163).

• As from the point above, it is unclear why the Adherence-VAS (Line 163-165) was not explained to the same detail as the VAS in the section above (Line 155-162). 

Response: We propose the following paragraph: “A 0 to 100 mm scale was used to assess adherence, with 100 mm indicating the poorest adherence with RA-related overall treatment and located at the right end, meanwhile 0 mm was located at the left end that corresponded to the best adherence with RA-related overall treatment” (Lines 178-181). 

• On Line 173 the authors name the section RP Questionnaire (RPQ). As this is the name of a tool used, spelling it out instead of putting an acronym on an acronym may be better.

Response: We have adopted the suggestion (Line 189).

• On Line 247-248, it would be helpful to have some indication on the type of patient who declined an interview seeing as they consisted 7.5% of the total sample.

Response: We agree with the reviewer; unfortunately we do not have patients’ characteristics but they were ambulatory patients, who declined based on the lack of time to commit with the interview. 

• In Table 2 (Line 265) and Table 3 (Line 295), RP is defined as risk score. For sake of consistency, suggest changing to “risk perception score” (Lines 283, 314).

Response: We have adopted the suggestion. 

• On Line 288 the authors refer to a PR score, which is assumed to actually be RP given the rest of the manuscript.

Response: The RP score is different to significant RP as highlighted in the “Material and methods” section, where the following sentence states “The RPQ score ranges from 0 to 100 mm, where 100 indicates the highest risk perception. Patients with a score ≥ 61.7 mm were considered to have significant RP [31]”. 

• On Lines 300-302, the authors indicate Table 4 shows “significant RP was associated with CAM use, female sex, years of formal education …”. This seems to imply the testing of RP as the dependent variable instead of CAM. This does not seem to align with the title of Table 4 which appears to imply CAM was the dependent variable.

Response: The reviewer is right, we have updated the sentence (Lines 320-322). 

• On Line 423 the authors use the word “finally” and then a sentence later (Line 430) they use “finally” again. Suggest a different word on Line 423.

Response: We suggest to star the sentence as” It needs to be highlighted that the association found between CAM use and access…” (Lines 442-443).

• On Line 448 the authors refer to the agent “thunder god wine”. This may be interpreted as a colloquial term for this treatment. This reviewer suggests maybe referring to it by its scientific name “Tripterygium wilfordii (thunder god wine)”.

Response: We have adopted the reviewer suggestion (Line 468).

• On Line 482-484 the authors refer to depression and health literacy being associated with CAM use. While they did use education years as a potential surrogate, maybe an explanation as to why this was not investigated may be prudent.

Response: We became aware of the association between health literacy and CAM use when a more exhaustive literature review was performed and a significant proportion of the sample size target was achieved. 

Reviewer #3:

1. Introduction. Paragraph 2, lines 84-86

What references support the statement of short-term benefits? Please insert the references that support this information.

2. Care must be taken when asserting short-term effectiveness using the reference provided. It reported paucity of randomized controlled trials in the area. The quality of the evidence does not allow to affirm the presence/absence of effectiveness of the intervention.

Response: We have updated the paragraph and added a reference that better supports our statement (Lines 87-90). 

3. Methods. Adherence-VAS, line 163

Were the evaluating physicians present in the treatment of patients?

If so, wouldn't that be a factor that would induce the patient's response?

Response: The primary rheumatologist was not present during the study assessments. We have updated the following section: “Interviews were performed immediately after patients visited their primary rheumatologists, in a different location within the outpatient clinic that was suitable for clinical research. The primary rheumatologist was not present during interview and study assessments” (Lines 151-154). 

4. Please indicate if the evaluating physicians were the same responsible for the care of the participants. If so, present it as a limitation of the study.

Response: They were not. We have updated the section as follows: “Interviews were performed immediately after patients visited their primary rheumatologists, in a different location within the outpatient clinic that was suitable for clinical research. The primary rheumatologist was not present during interview and study assessments” (Lines 151-154). 

5. Patient assessments, line 120. Did the evaluators show inter- and intra-evaluator reliability for the application of the evaluation instruments? Training alone does not guarantee the evaluator's reliability.

Please provide the evaluator's reliability data.

If reliability does not exist, present it as a limitation of the study.

Response: Repeatability (intra-observer) and reproducibility (inter-observer) were not examined; we have added as a limitation of the study (Lines 511-513).

6. Results. Paragraph 1, line 247. 266 participants were registering. However, the sample size was estimated at 220. Please explain the reason for registering extra participants.

Primary objective: the impact of RP on CAM use and additional associated factors

Response: When defining the sample size, CAM prevalence and its distribution in the patients with and without RP was considered. Early during the study development, it became evident that CAM use was highly prevalent in our population; in order to better balance CAM users between patients with and without RP, we ought to continue including patients (Importantly, considering study assessments, human and economic resources assigned to the study were in the low range). Finally, the number of patients enrolled was updated on weekly bases, instead of daily bases, which ended in a higher number of patients included. 

7. Paragraph 2, lines 84-86. The authors state that participants in the first group "ended to have more access to the federal health care system, lower Charlson scores, more frequent use of corticosteroids, higher scores on the pain-VAS, and more significant RP". However, even though the absolute numbers were different, there was no statistical difference between the groups. In this sense, there is statistical equality between the two groups.

Please consider this information

Response: The paragraph presents results that were statistically significant, but we also mentioned those were a statistical tendency was seen (“ They also tended to have more access to …..”); both groups of variables were ultimately included in the logistic regression analysis due to their statistical significance settled at p≤0.10 and/or their clinical relevance, as state in the corresponding section (Lines 293-295, 308-309).

8. Paragraph 4, line 288. The authors claim that the participants in the first group had higher scores for more significant PR. However, even though the absolute numbers were different, there was no statistical difference between the groups. Consider the previous comment.

Response: Please refer to previous comment. 

9. Discussion. General. Authors can go more directly to the focus. Concepts such as PR can be better addressed in the introduction. In this way, there will be more space for important discussions such as: difference in results when prayer was excluded from CAM; Mechanisms by which PR can influence CAM use.

Response: We have updated the Introduction and discussion section according to reviewer’s suggestion (Lines 95-115, 416-426). 

10, Paragraph 2. I believe that this paragraph should have a main focus on the possible mechanism by which PR influences CAM. I believe that conceptual terms of PR can be better addressed in the introduction.

Response: We have updated the discussion according to reviewer´s suggestion (Lines 392-415). 

11. Paragraph 6. Use a reference to support the statement:

“Patients primarily rely on their social networks for information regarding CAM strategies and are willing to try therapies even without the approval of their physicians.”

Response: We have provided 3 additional references (Lines 478-480).

12. Conclusion. The conclusion can go more directly to the study's findings. Previous study information is covered in the discussion.

Response: We have updated the conclusion according to the reviewer suggestion (Lines 515-520).

---

## [Decision Letter · Decision Letter 1]

21 Jul 2020

PONE-D-20-01920R1

Association of risk perception with the use of complementary and alternative medicine: a cross-sectional study in Hispanic patients with rheumatoid arthritis

PLOS ONE

Dear Dr. Pascual-Ramos,

Thank you for submitting your manuscript to PLOS ONE. After careful consideration, we feel that it has merit but does not fully meet PLOS ONE’s publication criteria as it currently stands. Therefore, we invite you to submit a revised version of the manuscript that addresses the points raised during the review process.

We look forward to receiving your revised manuscript.

Kind regards,

Daniel Steffens, PhD

Academic Editor

PLOS ONE

Reviewers' comments:

Reviewer's Responses to Questions

**Comments to the Author**

1. If the authors have adequately addressed your comments raised in a previous round of review and you feel that this manuscript is now acceptable for publication, you may indicate that here to bypass the “Comments to the Author” section, enter your conflict of interest statement in the “Confidential to Editor” section, and submit your "Accept" recommendation.

Reviewer #2: (No Response)

2. Is the manuscript technically sound, and do the data support the conclusions?

Reviewer #2: Yes

3. Has the statistical analysis been performed appropriately and rigorously? 

Reviewer #2: Yes

4. Have the authors made all data underlying the findings in their manuscript fully available?

Reviewer #2: Yes

5. Is the manuscript presented in an intelligible fashion and written in standard English?

Reviewer #2: Yes

6. Review Comments to the Author

Reviewer #2: Again, this reviewer believes this to be an interesting paper and stands by previous comments of its contribution to the literature. The authors appear to have fairly responded to and/or clarified all but one point requested. Below point not responded to and then one further comment:

1) Authors have not quite responded to what "PR" is (now on line 307). Is this a spelling mistake, because this reviewer cannot find where the acronym "PR" is defined? If the authors are saying "PR" = "significant RP", then this must be defined at first reference to this in the manuscript. This reviewer would advise against this as that would be extremely confusing and would request that it continues to be spelled out as "significant RP".

2) The authors have also added appendices for the scores they used. This is extremely helpful in understanding what was asked of patients. However, while Appendix 1 has both Spanish and English translation, Appendix 2 (I-CAM-Q) appears to only be in Spanish. If possible, it would be helpful to have the English translation for this as well. This reviewer defers to the Editor if this needs to be addressed.

Reviewer #3: All comments were answered. The mistakes were also clarified. Thus, I believe that the authors met the requirements for publication.

7. PLOS authors have the option to publish the peer review history of their article (what does this mean?). If published, this will include your full peer review and any attached files.

Reviewer #2: **Yes: **Sascha Karunaratne

---

## [Author Response · Author response to Decision Letter 1]

22 Jul 2020

Responses to reviewers

“Association of significant risk perception with the use of complementary and alternative medicine: a cross-sectional study in Hispanic patients with rheumatoid arthritis” (PONE-D-20-01920)

Authors’ responses 

“Association of risk perception with the use of complementary and alternative medicine: a cross-sectional study in Hispanic patients with rheumatoid arthritis (PONE-D-20-01920)”.

Reviewer #2: Again, this reviewer believes this to be an interesting paper and stands by previous comments of its contribution to the literature. The authors appear to have fairly responded to and/or clarified all but one point requested. Below point not responded to and then one further comment:

1) Authors have not quite responded to what "PR" is (now on line 307). Is this a spelling mistake, because this reviewer cannot find where the acronym "PR" is defined? If the authors are saying "PR" = "significant RP", then this must be defined at first reference to this in the manuscript. This reviewer would advise against this as that would be extremely confusing and would request that it continues to be spelled out as "significant RP".

Response: We apologize, it is a spelling mistake. We have reviewed the document and being particularly consistent with terms and acronyms.

2) The authors have also added appendices for the scores they used. This is extremely helpful in understanding what was asked of patients. However, while Appendix 1 has both Spanish and English translation, Appendix 2 (I-CAM-Q) appears to only be in Spanish. If possible, it would be helpful to have the English translation for this as well. This reviewer defers to the Editor if this needs to be addressed.

Response: We have included in Appendix 2 the International CAM questionnaire, which has a similar structure, although a different format, to the Spanish version that was used in the study. 

Reviewer #3: All comments were answered. The mistakes were also clarified. Thus, I believe that the authors met the requirements for publication.

Response: We appreciate the comment.

---

## [Editor Report · Decision Letter 2]

29 Jul 2020

Association of significant risk perception with the use of complementary and alternative medicine: a cross-sectional study in Hispanic patients with rheumatoid arthritis.

PONE-D-20-01920R2

Dear Dr. Pascual-Ramos,

We’re pleased to inform you that your manuscript has been judged scientifically suitable for publication and will be formally accepted for publication once it meets all outstanding technical requirements.

Kind regards,

Daniel Steffens, PhD

Academic Editor

PLOS ONE

---

## [Editor Report · Acceptance letter]

3 Aug 2020

PONE-D-20-01920R2 

Association of significant risk perception with the use of complementary and alternative medicine: a cross-sectional study in Hispanic patients with rheumatoid arthritis. 

Dear Dr. Pascual-Ramos:

I'm pleased to inform you that your manuscript has been deemed suitable for publication in PLOS ONE. Congratulations! Your manuscript is now with our production department. 

Kind regards, 

on behalf of

Dr. Daniel Steffens 

Academic Editor

PLOS ONE